# PADRe: A Unifying Polynomial Attention Drop-in Replacement for Efficient Vision Transformer

**Pierre-David Letourneau**[*]    **Manish Kumar Singh**[*]    **Hsin-Pai Cheng**
**Shizhong Han**    **Yunxiao Shi**    **Dalton Jones**    **Matthew Harper Langston**
**Hong Cai**    **Fatih Porikli**
Qualcomm AI Research[†]

`{pletourn,masi,hsinpaic,shizhan,yunxshi,daltjone,hlangsto,`
` hongcai,fporikli}@qti.qualcomm.com`

## ABSTRACT

We present Polynomial Attention Drop-in Replacement (**PADRe**), a novel and unifying framework designed to replace the conventional self-attention mechanism in transformer models. Notably, several recent alternative attention mechanisms, including Hyena, Mamba, SimA, Conv2Former, and Castling-ViT, can be viewed as specific instances of our PADRe framework. PADRe leverages polynomial functions and draws upon established results from approximation theory, enhancing computational efficiency without compromising accuracy. PADRe's key components include multiplicative nonlinearities, which we implement using straightforward, hardware-friendly operations such as Hadamard products, incurring only linear computational and memory costs. PADRe further avoids the need for using complex functions such as Softmax, yet it maintains comparable or superior accuracy compared to traditional self-attention. We assess the effectiveness of PADRe as a drop-in replacement for self-attention across diverse computer vision tasks. These tasks include image classification, image-based 2D object detection, and 3D point cloud object detection. Empirical results demonstrate that PADRe runs significantly faster than the conventional self-attention ($11\times \sim 43\times$ faster on server GPU and mobile NPU) while maintaining similar accuracy when substituting self-attention in the transformer models.

## 1 INTRODUCTION

Transformers have been pivotal in recent advancements across natural language processing, computer vision, and speech processing. In computer vision, Vision Transformers (ViTs) [9] are particularly significant, driving innovations in areas such as open-world recognition [15], image and video generation [25; 3], and large multi-modal models [19; 39].

However, the computational and memory costs associated with self-attention operations in transformers can increase quadratically relative to input size. This demanding challenge is particularly pronounced in real-world computer vision applications, which often involve inherently large input sizes, such as high-resolution images and videos in virtual reality and camera applications, or large 3D point clouds in autonomous driving. The need to run transformers on hardware platforms with limited computational and memory resources further intensifies these challenges.

To address these computational hurdles, we propose PADRe, a unifying Polynomial Attention Drop-in Replacement scheme. PADRe is a novel approach that can effectively substitute standard self-attention at low (linear computational and memory) costs without sacrificing accuracy. PADRe's unifying framework provides a guide for designing alternative transformer architectures. In particular, we observe and demonstrate that many recently-proposed attention replacement mechanisms (e.g., Hyena [24], Mamba [11], Conv2Former [14], SimA [16], Castling-ViT [41]), as well as the standard

---

[*]Equal contribution

[†]Qualcomm AI Research is an initiative of Qualcomm Technologies, Inc.

attention itself [35], may in fact be interpreted as specific instances of the PADRe framework (See Table 1).

Our framework relies on polynomial approximants in the input to replace the standard self-attention mechanism. Polynomials are well-studied objects known to be powerful and efficient multi-dimensional function approximants over compact sets [33]. We use these existing results, together with the aforementioned observations, as guiding principles. We also leverage simple, mobile-friendly operations, such as Hadamard products, to induce nonlinearities, providing the necessary modeling capacity of the polynomial functions. Moreover, our design allows for easily and efficiently scaling up the modeling capacity of existing network architectures by increasing the degree, while maintaining linear computational complexity and memory cost.

In addition to discussing the theoretical underpinnings and unifying properties of PADRe, we further experimentally demonstrate the significant advantages of PADRe over the standard self-attention through a variety of experiments involving computer vision tasks in practical contexts. More specifically, by replacing the self-attention operation with our design, we maintain the model performance while only incurring linear computational and memory costs. Further, by profiling on actual hardware platforms, we demonstrate the significantly better on-device efficiency of PADRe as compared to the standard self-attention operation.

We summarize the main contributions of the paper as follows:

- We propose PADRe, a unifying framework for replacing the attention mechanism based on polynomial approximants, which provides an efficient, linear-complexity alternative to the standard self-attention without sacrificing accuracy.

- We show that PADRe provides a unifying formulation for many of the recently-proposed attention replacement schemes, including Hyena [24], Mamba [11], Conv2Former [14], SimA [16], and Castling-ViT [41].

- We implement a specific instance of PADRe following the proposed design principles. We demonstrate the performance and advantage of PADRe through multiple experiments related to diverse, practical computer vision applications. Our PADRe-based models achieve similar or better accuracy than the original ones, while incurring significantly lower costs. Our on-device profiling further validates the computational efficiency of PADRe.

## 2 RELATED WORK

**Efficient Vision Transformer Backbones:** A majority of existing research proposes holistically designed vision transformer backbones. These backbones are capable of extracting features for various vision tasks, such as image classification, object detection, and segmentation. Most of these methods employ a hierarchical architecture, where the spatial dimension of the image feature map is progressively downsampled [20]. This strategy allows the deeper transformer layers to operate on smaller inputs, thereby reducing computational and memory costs. Specifically, many recent designs adopt convolution layers in the early stages of the network and apply transformers only to significantly downsampled feature maps later in the network [17; 13; 34; 38]. While this approach considerably improves efficiency, it still suffers from the inherent quadratic complexity when input size becomes large. Building upon hierarchical and/or convolution-transformer hybrid architectures, some studies propose more efficient, alternative attention schemes, e.g., ReLU-based attention [5], transposed attention [22], convolutional modulation [14], additive attention [27], shift-add attention [40], linear-angular attention [41]. However, all these models are designed to process a single 2D image and cannot be directly applied to more complex visual inputs, such as 3D point clouds. Many of them also rely on jointly designing the entire backbone to achieve good performance on 2D tasks and do not work well as drop-in replacements. In fact, using some of these efficient attention methods as drop-ins can degrade the model's performance, as reported by [12]. Another line of works study how to use fewer tokens to reduce computation while still utilizing the standard self-attention operation, e.g., [1; 29].

**Efficient Attention Drop-in Replacements:** To enhance the computational efficiency of vision transformers, various alternative attention mechanisms have been introduced. For instance, Swin [20] proposes windowed attention, where self-attention computation is limited to local neighborhoods of an image. Although this theoretically reduces computation, the intricate reshaping and indexing operations pose practical challenges on resource-constrained hardware platforms. Several works

propose linear-cost attentions [16; 41; 12; 27]. Some of them (e.g., [16; 12]) need to compute normalization factors at inference, which requires summing over all the tokens. This is inefficient on memory-constrained mobile platforms, especially when the number of tokens is large. Others (e.g., [41; 27]) are special cases of our proposed PADRe framework (see Section 3.5). Furthermore, it is worth noting that many of these alternative attentions still require multiple heads to maintain accuracy, a requirement that PADRe eliminates.

## 3 PADRe FRAMEWORK APPROACH

We describe our framework's technical details; at a high level, PADRe may be described as follows:

*Each element of the output tensor is a polynomial function of the elements of the input tensor, the coefficients of which are polynomial functions of the parameters (weights). Further, this dependency is such that the coefficients and the output itself may be computed efficiently (e.g., in linear time).*

For $N \in \mathbb{N}$ tokens and embedding / feature channel dimension $D \in \mathbb{N}$, consider an input $X \in \mathbb{R}^{N \times D}$. We denote $d$ as the *degree* of our framework; this parameter determines the strengths of the nonlinearities as discussed below. PADRe operations are performed on each batch element identically.

We omit batch dimension for brevity and consider a single head; if more than one head is used, each is treated independently using copies of the framework. PADRe consists of three main components: **(1) Linear transformations; (2) Nonlinearities; (3) Optional operations (e.g., output resizing, normalization)**. Further details regarding each step are discussed below in the following sections.

### 3.1 LINEAR TRANSFORMATIONS

Linear transformations consist of the following operations: for $i = 1, ..., d$, compute

$$Y_i = A_i X B_i, \tag{1}$$

where $A_i$ and $B_i$ are $N \times N$ and $D \times D$ weight matrices, and $Y_i \in \mathbb{R}^{N \times D}$. **We impose *structure* on the matrices $A_i$ and $B_i$ for efficient multiplication**; common structures include convolutions, sparse patterns, low-rank and hierarchical matrices, which can be optimized and parallelized on existing platforms. Note that left and right multiplications create a form of *mixing* among tokens (in the spatial dimension in the case of visual input) and in the embedding/channel dimension, respectively. Coupled with nonlinearities (Section 3.2) this leads to the presence of *cross-terms* (i.e., multivariate monomials) in the expansion that are key to the expressivity of the polynomial functions.

### 3.2 NONLINEARITIES

We construct nonlinear, polynomial functions of the input by first defining the matrices:

$$Z_1 = Y_1, \tag{2}$$
$$Z_{i+1} = (C_i Z_i D_i) \odot Y_{i+1}, \quad \text{for } i \in \{1, ..., d-1\}, \tag{3}$$

where $\odot$ indicates the Hadamard (element-wise) product, which is an efficient, parallelizable, and hardware-friendly operation.[†] $C_i$ and $D_i$ are $N \times N$ and $D \times D$ weight matrices that can perform additional token and channel mixing on the $Z_i$ tensor as needed, and $Z_i \in \mathbb{R}^{N \times D}$. Similarly, we require $C_i$ and $D_i$ to have structures that allow for efficient multiplications. In particular, we have the following lemma which is important for the analysis of the scheme,

**Lemma 1.** *The elements of $Z_i$ are homogeneous polynomials of degree $i$ in the input $X$.*

*Proof.* We proceed by induction. In the case $i = 1$,

$$Y_1 = A_1 X B_1, \tag{4}$$

which shows that each element of $Y_1$ is a linear function of the input. Now assume the elements of $Z_i$ are homogeneous polynomials of degree $i$ in the input and consider

$$Z_{i+1} = (C_i Z_i D_i) \odot (A_{i+1} X B_{i+1}). \tag{5}$$

---

[†]This is technically not the only possible choice; other polynomial bases, e.g., Chebyshev, Legendre, could be used for instance.

Observe that the elements of $A_{i+1} \, X \, B_{i+1}$ are homogeneous polynomials of degree 1 by construction and the elements of $C_i \, Z_i \, D_i$ are homogeneous polynomials of degree $i$ by assumption and construction (linear transformations do not affect this conclusion). The expression is therefore a homogeneous polynomial of degree $i + 1$ as claimed. $\qquad\square$

Next, we compute dense (rather than homogeneous) polynomials of the input to increase diversity, expressivity, and number of degrees of freedom, through the computation of the output tensor $P$,

$$[P]_{m,n} = \sum_{i=1}^{d} [W]_{m,n,i} \, [Z_i]_{m,n} + [L]_{m,n}, \qquad (6)$$

where $W \in \mathbb{R}^{N \times D \times d}$ is a weight tensor, $1 \leq m \leq N$ and $1 \leq n \leq D$ are token and embedding dimensions respectively, and $L \in \mathbb{R}^{N \times D}$ is the bias (i.e., $0^{\text{th}}$-order term).

### 3.3 Optional Operations

When an output of size different from to the input is required, we perform a computation of the form,

$$O = U \, P \, V, \qquad (7)$$

where $U \in \mathbb{R}^{F \times N}$, $V \in \mathbb{R}^{D \times G}$, and $O \in \mathbb{R}^{F \times G}$ (required output shape). As discussed, it is necessary that $U$ and $V$ possess structures (e.g., low-rank) to allow efficient application to $P$.

In addition, the $Y_i$'s tensors can optionally be normalized. While this is not necessary for the PADRe instances in our experiments, it can be a potentially useful regularization technique.

### 3.4 Overall Framework

Combining all operations together and leveraging Lemma 1, the output takes the following form:[†]

$$[P]_{m,n} = \sum_{i=1}^{d} [W]_{m,n,i} \, [Z_i]_{m,n} + [L]_{m,n} \sim \sum_{k \in \mathbb{N}^{N \times D} : |k| \leq d} \pi_k \, x^k, \qquad (8)$$

where $k$ is a multi-index, $|k| = \sum_{m=1}^{N} \sum_{n=1}^{D} k_{m,n}$ is the absolute degree, $\{\pi_k\}$ are coefficients (which have a complex polynomial relationship with the parameters/weights), and $x^k = \prod_{m=1}^{N} \prod_{n=1}^{D} x_{m,n}^{k_{m,n}}$ are monomials of the input variables. Equation 8 is a degree-$d$ multi-dimensional polynomial in the entries of the input $X$; that is, PADRe uses hardware-friendly and efficient operations to produce an output tensor whose entries are (potentially normalized) polynomials of the entries of the input tensor. In particular, any proposed scheme that can be written in this form falls within the PADRe framework.

### 3.5 Unifying Framework

Equation 8 provides a general, quantitative representation of PADRe's output. This formula in fact encompasses many recently-proposed schemes. Such observation may not be obvious at first sight but becomes clear following basic algebraic manipulations. For several recently proposed methods, we summarize their PADRe equivalence in Table 1 and provide detailed proofs to show that they are within our proposed PADRe framework in Appendix B. Further, it can be easily verified that the "attention" part of existing models based on the MetaFormer architecture [42] and linear token mixing (e.g., convolution [21], MLP [31; 30]) are also degree-1 PADRe equivalences.

While we focus on leveraging polynomial functions, PADRe can be generalized to rational functions (i.e., ratios of polynomials) as discussed in Appendix A.2. In addition, although the standard self-attention cannot be represented by polynomials, it can be approximated by high-degree rational functions, as we show in Appendix B.6.

---

[†]We ignore the optional resizing here, which is just a linear transformation.

| Attention Schemes | PADRe Equivalence | Proofs |
|---|---|---|
| SimA [16] | Normalized, homogeneous degree-3 polynomial | B.1 |
| Conv2Former [14] | Homogeneous degree-2 polynomial | B.2 |
| Hyena [24] | Homogeneous degree-$N$ polynomial (order-$N$ operator, commonly 3) | B.3 |
| Mamba [11] | Homogeneous degree-3 polynomial (approx.) | B.4 |
| Castling-ViT [41] | Degree-3 polynomial | B.5 |
| Self-attention [35] | High-degree rational function/ratio of polynomials (approx.) | B.6 |

Table 1: These existing attention schemes can be shown to be PADRe equivalences or can be approximated by PADRe. Detailed proofs can be found in Appendix B.

### 3.6 COMPUTATIONAL CHARACTERISTICS

We briefly analyze the number of parameters and computational cost of constructing an output $P \in \mathbb{R}^{N \times D}$ given an input $X \in \mathbb{R}^{N \times D}$ using PADRe. First, let us consider the number of parameters. Under our structural assumptions, we find $\mathcal{O}(N \cdot d)$ parameters for weight matrices $\{A_i\}$ and $\{C_i\}$, $\mathcal{O}(D \cdot d)$ parameters for weight matrices $\{B_i\}$ and $\{D_i\}$, and $\mathcal{O}(N \cdot D \cdot d)$ for the tensor $W$. Specifically, by construction, the PADRe framework requires that matrices have sufficient structure for an application in linear time. This implies no more than a linear scaling of the parameters. This results in a total of

$$\mathcal{O}(N \cdot D \cdot d) \tag{9}$$

parameters. For a fixed degree $d$, this scales like $\mathcal{O}(N \cdot D)$ which is linear in the size of the input. Therefore, the *memory cost is linear* in the size of the input.

As for the computational cost, we find that computing all $Y_i$'s costs $\mathcal{O}(N \cdot D \cdot d)$ FLOPs using the assumed structure. Further, each quantity $Y_i$ may be computed in parallel (data parallelism). Then, computing all $Z_i$'s carries a cost of $\mathcal{O}(N \cdot D \cdot d)$ FLOPs, which are computed sequentially based on the current scheme. Finally, computing the output $P$ comes at a cost of $\mathcal{O}(N \cdot D \cdot d)$ FLOPs. In total, the cost of computing the output $P$ from the input $X$ is therefore,

$$\mathcal{O}(N \cdot D \cdot d), \tag{10}$$

which is also linear in the size of the input for a fixed degree $d$.

We note the scheme only leverages highly-efficient matrix operations and not complex functions (e.g., $\exp(\cdot)$, $\cos(\cdot)$), which are often expensive on hardware and can cause quantization complications.

## 4 EXPERIMENTS

In this section, we comprehensively evaluate PADRe as a drop-in replacement of the standard self-attention for computer vision tasks, in terms of both accuracy and efficiency. More specifically, we implement a specific instance of PADRe following the design principles laid out in Section 3, and use it to substitute the self-attention in transformer models in several vision applications, such as image classification, single-image object detection, and point cloud object detection. We then measure the latency of our PADRe instances on two compute platforms: a server GPU and a mobile Neural Processing Unit (NPU), and compare with the standard self-attention. We show that PADRe provides similar or better accuracy on all the tasks, while being significantly faster, especially when the number of input tokens is large.

### 4.1 IMPLEMENTATION OF PADRE

As illustrated in Fig. 1, given an input tensor $X$, we first apply channel mixing and token mixing operations to generate $Y_i$, for each $i \in \{1, ..., d\}$; these mixing operations correspond to the linear transformations in Eq. 1. More specifically, we implement channel mixing using pointwise convolutions or linear layers. Token mixing is implemented with convolutions applied in the token dimension. When the input possesses an inherent 2D structure (e.g., image features), we use 11x11 2D convolutions to mix the tokens, after reshaping them into a 2D image format. On the other hand, when the input does not present a dense 2D structure, such as sparse 3D voxels from a point cloud, we treat them as a sequence of tokens and use 1D convolutions with kernel size 11 for mixing.

Next, we compute $Z_i$ for each degree $i$ based on $Y_i$ and $Z_{i-1}$, as described in Eq. 3. Channel and token mixings are further applied to $Z_i$ before it is used to produce degree-$(i + 1)$ terms. Finally,

all the $Z_i$'s are added together to generate the output $P$. Note that unlike the standard multi-head attention and existing alternative attention methods, we use a single head in all our experiments.[†]

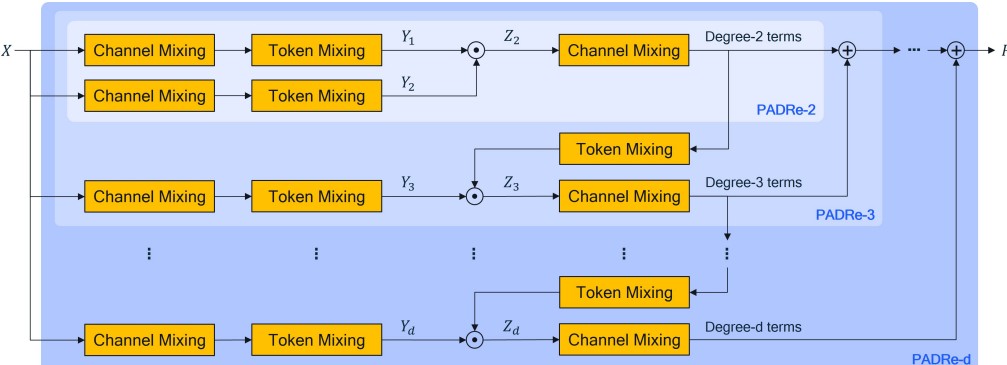

Figure 1: Our implementation of a PADRe-based attention drop-in replacement module. In our experiments, we use this to substitute the standard self-attention parts in existing transformer models. Note that this implementation approach is just a specific instance of our general PADRe framework.

In this specific instance, we do not have a degree-1 term in the summation that generates $P$. This is because there already exists a skip connection from $X$ to the output of the attention block in transformer, which provides a degree-1 term. In addition, we do not require an explicit degree-0 term (i.e., $L$ in Eq. 6) in the final summation, since the final convolutional layers provide the bias terms.

## 4.2 PERFORMANCE EVALUATION ON VISION APPLICATIONS

**Image Classification:** First, we consider the standard image classification task on ImageNet [26]. We use DeiT [32] as the baseline, including the tiny, small, and base network configurations, and replace the multi-head attentions with PADRe. We follow the default DeiT training setting (300 epochs without distillation) to train our models. Each training experiment takes ∼2 days on 8 NVIDIA A100 GPUs. We use the top-1 accuracy to measure classification performance.

Table 2 summarizes the results. It can be seen that by replacing self-attention with our PADRe instance, we achieve similar or better classification performance. Further, PADRe can run significantly faster on actual hardware platforms and scales much more efficiently w.r.t. increasing number of tokens (see Section 4.3).

Table 2 also compares PADRe with several recently proposed alternative attention methods, using DeiT baseline and the default DeiT training procedure for a fair comparison. We see that several proposed methods, such as Hydra attention [2] and linear-angular attention [41], fail to maintain accuracy. Other methods do improve accuracy, but are not as efficient as PADRe on actual compute platforms (Section 4.3).

**Single-Image Object Detection:** In this part, we consider a commonly used, end-to-end transformer-based object detector: DETR [6]. DETR consists of three main components: a ResNet backbone for image feature extraction, a transformer encoder with 6 (self-attention) layers, and a transformer decoder with 6 (self- and cross-attention) layers. In this application, we replace the self-attention in the 6 encoder layers with our PADRe instance. We train and evaluate our DETR with PADRe replacement on the COCO detection benchmark [18]. We follow the short-schedule [6] to train the model for 300 epochs on 8 NVIDIA A100 GPUs, which takes ∼4 days. We use the standard metric of Average Precision (AP) to measure detection performance.[†]

The results in Table 3 show that by substituting the standard self-attention with PADRe in the transformer encoder layers, we achieve a similar detection accuracy. In the encoder part, our PADRe instance also significantly reduces the computational cost by 55%.

---

[†]While using multiple heads is possible and straightforward to implement, we found that a single head was sufficient for PADRe to match the accuracy of multi-head attention models.

[†]See `https://cocodataset.org/#detection-eval` for more details on COCO object detection metrics.

| Baselines | Attention Mechanisms | Top-1 Accuracy↑ | FLOPs (G) | #Params (M) |
|---|---|---|---|---|
| DeiT-Tiny [32] | Standard self-attention [9] | 72.2 | 1.3 | 5 |
| | Hydra attention [2] | 68.3 | 1.1 | 6 |
| | Efficient attention [28] | 70.2 | 1.1 | 6 |
| | Linear-angular attention [41] | 70.8 | 1.1 | 6 |
| | Enhanced linear attention [5] | 72.9 | 1.1 | 6 |
| | k-NN attention [37] | 73.0 | 1.3 | 6 |
| | Focused linear attention [12] | 74.1 | 1.1 | 6 |
| | Vision Mamba [43] | 76.1 | 2.6 | 7 |
| | PADRe-2 (ours) | 74.5 | 1.2 | 5 |
| | PADRe-3 (ours) | 76.5 | 1.7 | 7 |
| DeiT-Small [32] | Standard self-attention [9] | 79.8 | 4.6 | 22 |
| | SimA [16] | 79.8 | 4.6 | 22 |
| | k-NN attention [37] | 80.1 | 4.6 | 22 |
| | Vision Mamba [43] | 80.5 | 9.6 | 26 |
| | PADRe-2 (ours) | 80.4 | 4.8 | 20 |
| | PADRe-3 (ours) | 80.5 | 6.1 | 26 |
| DeiT-Base [32] | Standard self-attention [9] | 81.8 | 17.6 | 86 |
| | Hydra attention [2] | 76.4 | 16.9 | - |
| | Hyena [24] | 78.5 | - | 87 |
| | PADRe-2 (ours) | 81.4 | 18.8 | 80 |

Table 2: Results on ImageNet-1K classification. By replacing standard self-attention with our PADRe instance, we achieve similar or better accuracy when comparing to the DeiT baseline and recently proposed alternative attention methods.

| Model | Attention Mechanisms | AP↑ | FLOPs (G) | #Params (M) |
|---|---|---|---|---|
| DETR [6] | Standard self-attention [9] | 40.6 | 10.1 | 6.1 |
| | PADRe-2 (ours) | 40.2 | 4.5 | 2.7 |

Table 3: Results on COCO detection, using DETR as the baseline. We achieve a similar accuracy but with a considerably lower computational cost.

| Baseline | Attention Mechanisms | NDS↑ | mAP↑ | FLOPs (G) | #Params (M) |
|---|---|---|---|---|---|
| DSVT [36] | Standard self-attention [9] | 71.1 | 66.4 | 14.9 | 1.1 |
| | PADRe-2 (ours) | 71.1 | 66.0 | 10.7 | 0.9 |

Table 4: Results on nuScenes LiDAR-based 3D object detection. We achieve a similar accuracy with a significantly lower computational cost.

**Point Cloud Object Detection:** Next, we evaluate PADRe on a more complex vision application where the input is no longer a single image. In this case, the input is a 3D point cloud. We consider the 3D point cloud object detection task on nuScenes [4], which involves detecting vehicles and other traffic elements based on a LiDAR point cloud. We select a transformer-based 3D object detection model, DSVT [36], which achieves state-of-the-art performance on nuScenes. In DSVT, a 3D transformer backbone extracts point cloud features, after which the features are projected to the birds-eye-view and further processed by 2D networks and a final detection head. We replace the self-attention in the 3D backbone with PADRe and follow the same training procedure as in [36]; training takes ∼2 days on 4 NVIDIA V100 GPUs. We use standard nuScenes metrics to measure detection performance, including mean average precision (mAP) and nuScenes Detection Score (NDS); NDS is computed as a weighted sum of 6 detection metrics over 10 object classes (see [4] for more details on the metrics). As the 3D voxels are sparse and cannot be arranged in a dense 2D format, we treat them as a sequence and use 1D convolutions to perform token mixing.

Table 4 summarizes the 3D object detection results on nuScenes validation set. After replacing the self-attentions with PADRe, we achieve similar detection accuracy (in terms of both NDS and mAP) as compared to the original model. We then measure the computational costs of the 3D transformer backbones in the baseline and our model. It can be seen that our modified DSVT 3D backbone uses 28% fewer FLOPs while maintaining detection accuracy as compared to the baseline.

### 4.3 LATENCY AND MEMORY EVALUATION ON HARDWARE PLATFORMS

We evaluate and compare the on-device latencies of our PADRe instance and the standard self-attention. Our goal is to provide a practical comparison of computational efficiency across different

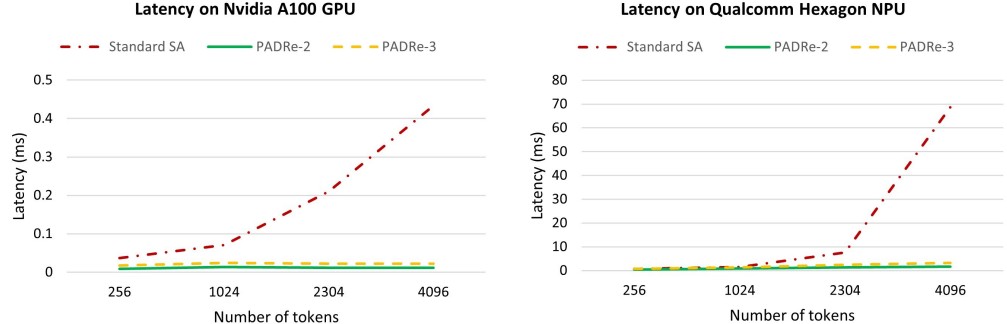

Figure 2: On-device (GPU and NPU) latency comparison for PADRe vs. standard self-attention. It can be observed that the latency of self-attention escalates significantly with an increase in the number of input tokens. In contrast, PADRe demonstrates a linear growth pattern.

hardware platforms. Specifically, we measure latencies on two compute platforms: 1) Nvidia A100 GPU, and 2) Qualcomm® Hexagon™ NPU (included in the Snapdragon® 8 Gen 3 processor on Samsung S24),[†] which is an AI accelerator specialized for neural network workloads. On the NPU, we quantize the network weights and activations to Int8 when measuring latency.

Since our objective is to replace the standard self-attention, we specifically evaluate only the "attention" components on-device. More specifically, for PADRe, we run the instance described in Section 4.1 and shown in Fig. 1, and for the standard transformer, we run a multi-head attention operation. Using the hyperparameters of DeiT-Tiny, we set the embedding dimension to 192 in both PADRe and self-attention, and use 3 heads in the multi-head attention. Note that we only use a single head in our PADRe instance in all our experiments. We evaluate on different numbers of input tokens, including: 256, 1024, 2304, and 4096, which correspond to input image resolutions of 256×256, 512×512, 768×768, 1024×1024, assuming a 16×16 patchification (as used in ViT).

Fig. 2 shows the on-device latencies w.r.t. number of tokens. We see that the standard attention mechanism exhibits a significant latency growth as the number of tokens increases. In contrast, our proposed PADRe maintains linear latency proportional to the number of tokens. This confirms that PADRe provides a significant efficiency advantage over self-attention when deployed on actual hardware, including large GPUs and resource-constrained mobile NPUs.

Furthermore, we evaluate the memory consumption. Considering one layer, given 4096 input tokens, PADRe's peak memory consumption is 106MB while that of self-attention is 145MB, 37% higher than PADRe. Such advantages are expected to become even more prominent as one increases the number of tokens, given the linear scaling of PADRe.

## 4.4 Ablation Study on Polynomial Degree in PADRe

In this study, we scale up the polynomial degree of our PADRe instance following the implementation in Section 4.1, and show its effect on the model performance and computational costs. Table 5 shows that by increasing the polynomial degree, the network obtains stronger modeling capacity, leading to performance improvements. Specifically, when using degree 3, PADRe has a significantly better accuracy over the baseline model, while being more computationally efficient on device. By further increasing the degree beyond 3, we observe additional small accuracy gains.

| Baseline | Attention Mechanisms | Top-1 Accuracy↑ |
|---|---|---|
| | Standard self-attention [9] | 72.2 |
| | PADRe-2 (ours) | 74.5 |
| DeiT-Tiny [32] | PADRe-3 (ours) | 76.5 |
| | PADRe-4 (ours) | 76.9 |

Table 5: ImageNet classification accuracy when scaling up PADRe degree.

Table 6 shows the computational costs of our PADRe instance with different degrees, and compares those with the standard self-attention and other recently-proposed efficient attention methods such

---

[†]Snapdragon and Qualcomm branded products are products of Qualcomm Technologies, Inc. and/or its subsidiaries. Qualcomm patented technologies are licensed by Qualcomm Incorporated.

| Attention Mechanisms | FLOPs (G) | Latency on GPU (ms) | Latency on NPU (ms) |
|---|---|---|---|
| Standard self-attention [9] | 1.31 | 0.43 | 68.60 |
| Linear-angular attention [41] | 1.22 | 0.53 | - |
| Focused linear attention [12] | 1.25 | 0.08 | - |
| PADRe-2 (ours) | 1.10 | 0.01 | 1.62 |
| PADRe-3 (ours) | 2.28 | 0.02 | 3.17 |
| PADRe-4 (ours) | 3.27 | 0.04 | 4.22 |

Table 6: FLOPs and on-device (both GPU and NPU) latency comparison of standard self-attention, recent efficient attention methods, and PADRe of various degrees.[†] The measurement is based on the attention parts only, with 4096 input tokens.

as linear-angular attention [41] and focused linear attention [12]. We follow the same setup in the previous section, where we set the channel to 192, use 3 heads in self-attention and existing efficient attentions, and 1 head in our PADRe instance. The number of input tokens is fixed to be 4096.

Increasing the degree in PADRe allows for efficiently scaling up model capacity and performance. Even with degree-4, PADRe remains significantly faster than self-attention, especially on the mobile NPU. While existing efficient attentions use fewer FLOPs, some can have higher latency on device. It is noteworthy that FLOPs is not an accurate indicator of model efficiency on actual hardware; similar observations have also been made in other works, e.g., [7]. For instance, operations that are not well optimized or cause memory bottlenecks on device are not reflected by FLOP count.

## 5 DISCUSSION

**Relation to Taylor Series:** One may consider directly approximating sufficiently regular nonlinearities, including the self-attention, using a Taylor expansion. We have performed such attempt, and have observed the accuracy to be greatly affected by small errors in the approximation, thus requiring an impractically large degree. Our PADRe framework, by contrast, learns a different nonlinearity without affecting the output or accuracy in any significant way. This more flexible framework, although inspired by Taylor's theorem, is not equivalent, and has been shown to be beneficial.

**Comparing with FlashAttention [8]:** FlashAttention reduces memory cost to $\mathcal{O}(N)$ by decomposing the softmax computation, but still requires $\mathcal{O}(N^2)$ computation. In addition, it is only designed for specific hardware, e.g., Nvidia A100 GPU. Our proposed PADRe, on the other hand, requires only $\mathcal{O}(N)$ memory and computation costs, and is generally applicable. Furthermore, PADRe is amenable to low-level optimization (e.g., tiling) for on-device deployment as it consists of hardware-friendly operations.

**Limitations and Future Work:** In this paper, we have extensively studied using PADRe to replace the standard self-attention and have shown that this is more computationally efficient and maintains model performance. However, we did not present empirical results of its application to cross-attention and/or multi-modal operations. In principle, PADRe can be extended for cross-attention, a discussion of which is provided in Appendix A.1; this would be interesting as part of future work.

We observe that model performance begins to saturate when scaling up the degree of our implemented PADRe instance beyond 3 (see Table 5). We have yet to establish the source of this phenomenon. One possibility may have to do with the numerical stability of monomials. In future work, we will investigate the use of different, more stable polynomial bases, such as orthogonal polynomials. In addition, depending on the task and data, it is possible to find the optimal degree via existing neural architecture search methods, e.g., [10].

While we mainly apply PADRe to computer vision tasks, it can be applied for any use case with transformer models. Specifically, PADRe can be easily extended to sequential applications like natural language processing and time series analysis. In a preliminary study, we used a reduced Llama2 with $\sim 300M$ parameters trained on SlimPajama as the baseline. Then, we replaced the self-attention with PADRe and retrained the model. On the test set, PADRe achieves similar cross-entropy. We will consider further investigations of LLMs as part of future work.

---

[†]We do not report the latencies of these alternative attention methods on NPU, as it would require significant effort to rewrite the implementation in order to run them on the mobile NPU.

## 6 CONCLUSIONS

In this paper, we proposed a unified, polynomial drop-in attention replacement framework, PADRe, which leverages polynomials to replace the expensive standard self-attention in transformers. More specifically, we presented the key components for constructing an expressive, efficient, and hardware-friendly scheme to substitute self-attention. In addition, PADRe is a general framework that encompasses many recently-proposed alternative attention schemes; we provided detailed proofs to show this. In order to demonstrate the efficacy of PADRe on practical applications and actual hardware platforms, we implemented a specific instance of PADRe based on the design principles, using efficient and hardware-friendly operations. Through extensive experiments, we showed that PADRe provides an efficient replacement of self-attention that maintains model performance and incurs much lower computational costs. Specifically, we showed that our PADRe instance runs significantly faster on both a power GPU (NVIDIA A100) and a resource-constrained mobile NPU (Qualcomm Hexagon tensor processor), by as much as $43\times$ when comparing to the standard self-attention.

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

## A  PADRe GENERALIZATIONS

In this section, we present a few generalization of the framework, namely, generalization to the cross-attention mechanism, as well as a rational version of the framework.

### A.1  MULTI-MODAL PADRe (I.E., CROSS-ATTENTION)

PADRe can be generalized to multi-modal input in many ways. We present one here. Imagine that $X^{(a)} \in \mathbb{R}^N \times D$ represents input from one mode and $X^{(b)} \in \mathbb{R}^{M \times D}$ represents input from another mode. Then, we can define $\hat{Y}_i^{(a)}$ and $\hat{Y}_i^{(b)}$ for each mode just as before (linear transformations should be chosen so that they are of equal sizes). To capture nonlinear interactions among modes, the difference lies in the computation of $\hat{Z}_j$. In this case, we write,

$$Z_j = \odot_{i=1}^{j} Y_i^{(c_i)}, \tag{11}$$

where $c_i \in \{a, b\}$ for each $i$. For instance, if the degree is $d = 3$ and $c = (a, a, b)$, then one finds that,

$$Z_j = Y_0^{(a)} \tag{12}$$

$$Z_j = Y_0^{(a)} \odot Y_1^{(a)} \tag{13}$$

$$Z_j = Y_0^{(a)} \odot Y_1^{(a)} \odot Y_2^{(b)}. \tag{14}$$

It is important for the sequence is not trivial, i.e., to contain cross-terms, not $(a, a, a)$ nor $(b, b, b)$. This will capture nonlinear interactions between modes. Also, note that more than two modes could be involved, and that multiple sequences may be considered.

### A.2  RATIONAL APPROXIMATION

This section describes a generalization of PADRe. Although we found that the standard PADRe based on polynomial approximations offered great performance in the context of computer vision, it is expected that certain applications will benefit from this more powerful framework. In particular, the following approach is inspired by the improved effectiveness of PADRe (rational) approximants compared with polynomial (Taylor) approximants.

The main difference is that the drop-in replacement takes the form of a *rational functions* rather than a *polynomial function*. As discussed in [23] for instance, in the one-dimensional case, the former have much higher approximation power than the latter. Further, from an intuitive perspective, adding flexibility to the denominator allows one to better capture "poles" (or areas of high variability) often observed when using `softmax`. Finally, the introduction of a function at the denominator might be seen as a replacement for the normalization generally used prior to the attention mechanism.

From a technical standpoint, we construct a degree-$[d/e]$ rational PADRe as follows: just as before, we let,

$$Y_i = A_i \, X \, B_i \tag{15}$$

for $i = 1, ..., d + e$, and,

$$\mathcal{K}_j = \odot_{i=1}^{j} \hat{Y}_i, \tag{16}$$

$$\mathcal{L}_k = \odot_{i=1}^{k} \hat{Y}_{d+i}, \tag{17}$$

for $j = 1, ..., d$ and $k = 1, ..., e$. Then, we define

$$\mathcal{N}_{a,b} = \sum_{j=1}^{d} W_{a,b,j} \, [\mathcal{K}_j]_{a,b} + V_{a,b} \tag{18}$$

$$\mathcal{D}_{a,b} = \sum_{k=1}^{e} Q_{a,b,j} \, [\mathcal{L}_j]_{a,b} + P_{a,b}. \tag{19}$$

Finally, the output (prior to resizing) takes the form (i.e., element-wise "Hadamard division"),

$$O_{a,b} = \frac{\mathcal{N}_{a,b}}{\mathcal{D}_{a,b}} \tag{20}$$

The cost of computing such an approximation amounts to that of computing the numerator and denominator. Since both are PADRe-like, we expect a cost of roughly $2\times$ that of the polynomial-only scheme, and thus linear scaling in the size of the input $N$.

Finally, note that care should be taken at training time to avoid a vanishing denominator (and ensuing numerical instability). This may be done by, for example, squaring the denominator and adding a small regularizing positive term.

## B    PADRe- SPECIAL CASES

### B.1    SIMA [16]

SimA (Simple Attention), as its name suggests, is a simple mechanism for replacing the standard attention. The output $O$ of SimA can be expressed as,

$$O = \hat{Q}\,\hat{K}^T\,V \tag{21}$$

where,

$$\hat{Q}^i = \frac{Q^i}{||Q^i||_1}, \; \hat{K}^i = \frac{K^i}{||K^i||_1}, \tag{22}$$

and $\cdot^i$ indicates the $i^{th}$ column of a matrix ($\ell_1$ column-wise normalization), and where,

$$Q = X\,W_Q, \; K = X\,W_K, \; V = X\,W_V, \tag{23}$$

for some appropriate projection matrices as usual. Substitution implies that,

$$O_{m,n} = \sum_{i=1}^{D} \hat{Q}_{m,i} \left( \sum_{j=1}^{N} \hat{K}_{j,i}\,V_{j,n} \right) \tag{24}$$

$$= \sum_{i=1}^{D} \frac{[X\,W_Q]_{m,i}}{||Q^i||_1} \left( \sum_{j=1}^{N} \frac{[X\,W_K]_{j,i}}{||K^i||_1} [X\,W_V]_{j,n} \right) \tag{25}$$

$$= \sum_{i=1}^{D} \sum_{j=1}^{N} \frac{1}{||Q^i||_1||K^i||_1} \left( \sum_{k_Q=1}^{D} \sum_{k_K=1}^{D} \sum_{k_V=1}^{D} \left([W_Q]_{k_Q,i}[W_K]_{k_K,i}\,[W_V]_{k_V,n}\right) \left(X_{m,k_Q}\,X_{j,k_K}\,X_{j,k_V}\right) \right) \tag{26}$$

This should be recognized as a homogeneous degree-3 polynomial with coefficients of the form,

$$\frac{\left([W_Q]_{k_Q,i}[W_K]_{k_K,i}\,[W_V]_{k_V,n}\right)}{||Q^i||_1||K^j||_1} \tag{27}$$

i.e., exhibiting a (normalized) polynomial relationship with the parameters/weights. Therefore, as claimed, SimA is a special case of the PADRe framework.

### B.2    CONV2FORMER [14]

Given an input $X \in \mathbb{R}^{H \times W \times C}$, Conv2Former replaces the self-attention mechanism with a block of the form,

$$Z = A \odot V \tag{28}$$
$$A = \mathrm{DConv}_{k \times k}\left(W_1\,X\right) \tag{29}$$
$$V = W_2\,X \tag{30}$$

where $Z$ is the output, $\odot$ represents the Hadamard product, $W_1$ and $W_2$ are weight matrices of two linear layers, and $\mathrm{DConv}_{k \times k}$ denotes a depthwise convolution with kernel size $k \times k$, which is really just an efficient linear transformation applied to each channel.

Letting $C$ be a tensor representing the convolution kernel and expanding formulas, we find that,

$$[\text{DConv}_{k \times k}(W_1 X)]_{i,j,k} = \sum_{p,q} C_{i-p,j-q,k} \left( \sum_{m,n} [W_1]_{(p,q),(m,n)} X_{m,n,k} \right) \tag{31}$$

$$[W_2 X]_{i,j,k} = \sum_{s,t} [W_2]_{(i,j),(s,t)} X_{s,t,k} \tag{32}$$

Therefore, it follows that,

$$[Z]_{i,j,k} = [A \odot V]_{i,j,k} = \left( \sum_{p,q} C_{i-p,j-q,k} \left( \sum_{m,n} [W_1]_{(p,q),(m,n)} X_{m,n,k} \right) \right) \cdot \left( \sum_{s,t} [W_2]_{(i,j),(s,t)} X_{s,t,k} \right) \tag{33}$$

$$= \sum_{m,n} \sum_{s,t} \left( [W_2]_{(i,j),(s,t)} \sum_{p,q} C_{i-p,j-q,k} [W_1]_{(p,q),(m,n)} \right) X_{m,n,k} X_{s,t,k} \tag{34}$$

This should be recognized as a degree-2 (homogeneous) polynomial in the input $X$, and therefore an specific instance of the PADRe framework.

### B.3 HYENA [24]

The proposed `Hyena` operator is defined as follows (Definition 3.1, [24]),

**Definition B.1.** (Order-$N$ Hyena Operator) *Let $(x^0, x^1, ..., x^N)$ be projections of the input and let $h^1, ..., h^N$ be a set of learnable filters. The Hyena-$N$ operator is defined by the recurrence:*

$$z_t^1 = x_t^0 \tag{35}$$

$$z_t^{n+1} = x_t^n (h^n * z^n)_t, \quad n = 1, ..., N \tag{36}$$

$$y_t = z_t^{N+1} \tag{37}$$

That is, Hyena proposes to use compositions of $N$ element-wise multiplications and convolutions as a drop-in replacement for the attention. Such operators possess these two important properties:

1. **Homogeneous polynomial functions**: the order-$N$ Hyena Operator corresponds to the evaluation of a degree-$(N+1)$ homogeneous polynomial (or equivalently $(N+1)$-tensor) which coefficients are non-linear functions of the parameters.[†]

2. **Fast evaluation**: the non-linear dependence of the coefficients (or equivalent the tensor core elements) on the parameters are such that the operator may be evaluated rapidly (i.e., in quasi-linear time).

More precisely, letting $\chi$ be the original input vector and $n := (n_0, n_1, ..., n_N) \in \mathbb{N}^{N+1}$ be multi-indices,[†] we argue that the order-$N$ Hyena operator may be written as,

$$y = \sum_n \eta_n \left( \prod_{i=0}^N \chi_{n_i} \right) \tag{38}$$

for some coefficients $\{\eta_n\}$ that are polynomial functions of the parameters ($\{h^i\}$ and linear projections). This is a a homogeneous polynomial of degree $(N+1)$. In light of these observations, one finds that *Hyena is in fact a special case of PADRe*.

We present analysis of the Order-$N$ Hyena Operator which shows it is a specific instances of PADRe. In particular, we argue that the latter really is a homogeneous polynomial of degree $(N+1)$, which

---

[†]Higher levels of non-linearity ($N$) increase expressivity, while the global nature of convolutions and projections ensures that all correlations among tokens are taken into account at multiple scales.

[†]We use $\chi$ in order to differentiate between the Hyena notation which uses $x$ as projections rather than input.

may also be interpreted as an $(N + 1)$-tensor (multilinear functional/operator). Following the above definition and Remark 3.1 ([24]), an order-$N$ Hyena operator may be expressed as,

$$y = x^N \cdot (h^N * (x^{N-1} \cdot (h^{N-1} * (...)))) \tag{39}$$

This expression can be expanded as follows,

$$y_{t_N} = x_{t_N}^N \sum_{t_{N-1}=0}^{L} h_{t_N-t_{N-1}}^N \left( x_{t_{N-1}}^N \sum_{t_{N-2}=0}^{L} h_{t_{N-1}-t_{N-2}}^{N-1} \left( ... \left( x_{t_1}^1 \sum_{t_0=0}^{L} h_{t_1-t_0}^1 x_{t_0}^0 \right) \right) \right) \tag{40}$$

Using commutativity to re-arrange the latter, we find that

$$y_{t_N} = \sum_{t_0=0}^{L} ... \sum_{t_{N-1}=0}^{L} \left( \prod_{i=1}^{N} h_{t_i-t_{i-1}}^i \right) \left( \prod_{i=0}^{N} x_{t_i}^i \right), \tag{41}$$

where $t_i \in \{0, ..., L\}$ for every $i$. Now, recall that each vector $x^i$ represents a projection of the input $\chi \in \mathbb{R}^M$ onto a $L$-dimensional space.[†] Let us write,

$$x_{t_i}^i = [P^i \chi]_{t_i} = \sum_{m_i=1}^{M} P_{t_i,m_i}^i \chi_{m_i}. \tag{42}$$

Upon substitution, we find that,

$$\prod_{i=0}^{N} x_{t_i}^i = \prod_{i=0}^{N} \left( \sum_{m_i=1}^{M} P_{t_i,m_i}^i \chi_{m_i} \right) \tag{43}$$

$$= \sum_{m_0=0}^{M} ... \sum_{m_N=0}^{M} \left( \prod_{j=0}^{N} P_{t_j,m_j}^j \right) \left( \prod_{j=0}^{N} \chi_{m_j} \right) \tag{44}$$

and therefore,

$$y_{t_N} = \sum_{t_0=0}^{L} ... \sum_{t_{N-1}=0}^{L} \left( \prod_{i=1}^{N} h_{t_i-t_{i-1}}^i \right) \left( \sum_{m_0=0}^{M} ... \sum_{m_N=0}^{M} \left( \prod_{j=0}^{N} P_{t_j,m_j}^j \right) \left( \prod_{j=0}^{N} \chi_{m_j} \right) \right) \tag{45}$$

$$= \sum_{m_0=0}^{M} ... \sum_{m_N=0}^{M} \left( \sum_{t_0=0}^{L} ... \sum_{t_{N-1}=0}^{L} \left( \prod_{i=1}^{N} h_{t_i-t_{i-1}}^i \prod_{j=0}^{N} P_{t_j,m_j}^j \right) \right) \left( \prod_{j=0}^{N} \chi_{m_j} \right) \tag{46}$$

$$= \sum_{m_0=0}^{M} ... \sum_{m_N=0}^{M} \eta_{(m_0,m_1,...,m_N)} \left( \prod_{j=0}^{N} \chi_{m_j} \right) \tag{47}$$

as claimed. The nonlinear dependence of the coefficients $\eta_m$ on the parameters $\{h^i\}_{i=1}^N$ and $\{P^j\}_{j=0}^N$ takes the form,

$$\eta_m = \sum_{t_0=0}^{L} ... \sum_{t_{N-1}=0}^{L} \left( \prod_{i=1}^{N} h_{t_i-t_{i-1}}^i \prod_{j=0}^{N} P_{t_j,m_j}^j \right) \tag{48}$$

$$= \sum_{t_0=0}^{L} ... \sum_{t_{N-1}=0}^{L} \left( P_{t_N,m_N}^N \left( \prod_{i=1}^{N} h_{t_i-t_{i-1}}^i P_{t_{i-1},m_{i-1}}^{i-1} \right) \right) \tag{49}$$

$$= P_{t_N,m_N}^N \prod_{i=1}^{N} \left( \sum_{t_{i-1}=0}^{L} h_{t_i-t_{i-1}}^i P_{t_{i-1},m_{i-1}}^{i-1} \right) \tag{50}$$

That is, the coefficients are products of convolutions applied to the projections in each dimension (with the exception of the last dimension which is a mere scalar multiplication). This demonstrates the claim.

---

[†] $M$ is used to represent the dimension of the input, whatever it is.

### B.4 Mamba [11]

In this section, we argue that the mamba SSM framework is in fact a special case of our PADRe framework. The mamba framework is a state-space model based on the following recursion:

$$h_t = \bar{A} h_{t-1} + \bar{B} x_t \tag{51}$$

$$y_t = C h_t \tag{52}$$

where $t$ is the timestep, $h_t$ is the hidden state at time $t$, $x_t$ is the input at time $t$ and $y_t$ is the output at time $t$. The "bars" above the matrices is mamba notation for discretization, which takes the form,

$$\bar{A} = e^{\Delta A} \tag{53}$$

$$\bar{B} = (\Delta A)^{-1} \left( e^{\Delta A} - I \right) \Delta B \tag{54}$$

where $\bar{A}$ is an $N \times N$ matrix chosen to be *diagonal*, $\bar{B}$ is an $N \times 1$ vector and $C$ is an $1 \times N$ vector.

We now discuss technical details. First, we have the following lemma,

**Lemma 2.** *If $h_0 = 0$, the discretized system of Equations 51 has solution,*

$$y_t = \sum_{n=0}^{t} \left( C \bar{A}^{t-n} \bar{B} \right) x_n \tag{55}$$

*Proof.* We proceed by induction. Clearly for $t = 0$, we get,

$$y_0 = C \bar{B} x_0 = C \bar{A}^0 \bar{B} x_0 \tag{56}$$

Now assume the induction hypothesis holds and consider,

$$y_{t+1} = C \left( \bar{A} h_t + \bar{B} x_{t+1} \right) \tag{57}$$

$$= C \left( \bar{A} \sum_{n=0}^{t} \left( \bar{A}^{t-n} \bar{B} \right) x_n + \bar{B} x_{t+1} \right) \tag{58}$$

$$= C \sum_{n=0}^{t+1} \left( \bar{A}^{t+1-n} \bar{B} \right) x_n \tag{59}$$

which proves the claim. $\qquad \square$

The main difference with regular SSM is that mamba allows $B$, $C$ and $\Delta$ to be functions of the input (Algorithm 2). Specifically, $B$ and $C$ are *linear* function of the input whereas $\Delta$ is the `softplus` of a linear function of the input, i.e.,

$$B = W_B x \tag{60}$$

$$C = x^T W_C \tag{61}$$

$$\Delta(x) = \frac{1}{\beta} \log \left( 1 + e^{\beta (\pi + x W_\Delta)} \right) \tag{62}$$

where $W_B$ is a $N \times L$ matrix, $W_C$ is a $L \times N$ matrix, $W_\Delta$ is a (rank-1) $D \times D$ matrix, $B$ and $C$ are size $N \times 1$ and $1 \times N$ vectors, respectively, and $\pi$ is some parameter.

For what follows, we make the assumption that,

$$0 \le \Delta(x) \ll 1 \tag{63}$$

uniformly in the input $x$. In this case, we find that,

$$\bar{A} = e^{\Delta A} = I + \Delta A + O(\Delta^2) \tag{64}$$

$$\bar{B} = (\Delta A)^{-1} \left( e^{\Delta A} - I \right) \Delta B \tag{65}$$

$$= A^{-1} \left( I + \Delta A + O(\Delta^2) - I \right) B \tag{66}$$

$$= \Delta B + O(\Delta^2) \tag{67}$$

Therefore, under our assumption, it is reasonable to expect that,

$$\bar{A} \approx I + \Delta A \tag{68}$$

$$\bar{B} \approx \Delta B = \Delta W_B x \tag{69}$$

$$C = x^T W_C \tag{70}$$

Substituting these expressions in the recursion, we find that,

$$y_t = C \sum_{n=0}^{t} \bar{A}^{t-n} \left( \bar{B} x_n \right) \tag{71}$$

$$\approx \Delta \left( x^T W_C \right) \left( \sum_{n=0}^{t} \bar{A}^{t-n} x_n \right) (W_B x) \tag{72}$$

$$= \sum_{n=0}^{t} \left( \sum_{i,j=1}^{L} \left[ W_C \bar{A}^{t-n} W_B \right]_{i,j} (x_i \cdot x_j \cdot x_n) \right) \tag{73}$$

$$= \sum_{n=0}^{t} \sum_{i,j=1}^{L} c_{i,j}^{(t,n)} (x_i \cdot x_j \cdot x_n) \tag{74}$$

where,

$$c_{i,j}^{(t,n)} := \left[ W_C \bar{A}^{t-n} W_B \right]_{i,j} \tag{75}$$

This should be recognized as a degree-3 homogeneous polynomial in the input $x$. Further, the coefficients $c_{i,j}^{(t,n)}$ exhibit a polynomial relationship in the parameters $(A, W_B, W_C)$ (assuming a truncated Taylor series for the exponential). Furthermore, the matrix $A$ exhibit such structures that the latter may be computed rapidly. Thus, it follows that Mamba is a special case of our PADRe framework.

## B.5 CASTLING-VIT [41]

In this section, we argue that Castling-ViT also fall within the bounds of the PADRe framework. Recall that at inference, the attention replacement used by Castling-ViT can be written as,

$$O = \frac{1}{\pi} Q \left( K^T V \right) + \left( \frac{1}{2} I + M_{DW} \right) V \tag{76}$$

where,

$$Q = X W_Q, \ K = X W_K, \ V = X W_V, \tag{77}$$

$I$ is the identity, and $M_{DW}$ represents a learnt depth-wise convolution (i.e., a fast linear transformation). Upon substitution, we find that the left-hand term in Eq.equation 76 is a degree-3 homogeneous polynomial in the input $X$, whereas the right-hand term is a linear function of the input. All in all, this implies that the output $O$ of the Castling ViT module is a degree-3 polynomial that can be computed rapidly (i.e., in linear time). It is therefore an instance of the PADRe framework.

## B.6 STANDARD ATTENTION [35]

In this section, we discuss the standard attention and argue that it may also be interpreted as an (approximate) instance of PADRe, albeit an instance of the *rational* framework (Section A.2) with a large degree (larger than practically acceptable).

To see how this is so, we write the attention explicitly as,

$$\left[ \texttt{Attention}(Q, K, V) \right]_{m,n} = \left[ \texttt{softmax} \left( \frac{Q K^T}{\sqrt{d_k}} \right) V \right]_{m,n} \tag{78}$$

$$= \sum_{i} \frac{e^{\left[ \frac{Q K^T}{\sqrt{d_k}} \right]_{m,i}}}{\sum_{j} e^{\left[ \frac{Q K^T}{\sqrt{d_k}} \right]_{m,j}}} V_{i,n} \tag{79}$$

where,

$$Q = X\,W_Q, \; K = X\,W_K, \; V = X\,W_V, \tag{80}$$

for some weight/projections matrices, as usual. Now comes the main observation: thanks to the normalization $\frac{1}{\sqrt{d_k}}$, one may expect that the elements $\frac{Q\,K^T}{\sqrt{d_k}}$ will generally lie in some canonical interval $[-L, L]$ for some $0 < L \in \mathbb{R}$. In this case, the exponential functions may be expanded using the Taylor series as,

$$e^{\left[\frac{Q\,K^T}{\sqrt{d_k}}\right]_{m,i}} = \sum_{l=0}^{d} \frac{\left[\frac{Q\,K^T}{\sqrt{d_k}}\right]_{m,i}^{l}}{l!} + O\left(\frac{L^{d+1}}{(d+1)!}\right) \tag{81}$$

for some degree $d$ sufficiently large that the error term is on the order of $0 < \epsilon \ll 1$. Note that such a truncated Taylor expansion may be assumed to be accurate over this fixed interval only.

Upon substitution and expansion, we therefore find that,

$$[\texttt{Attention}(Q, K, V)]_{m,n} \approx \frac{\sum_i \sum_{l=0}^{d} \frac{[Q\,K^T]_{m,i}^{l}}{d_k^{l/2}\, l!}\, V_{i,n}}{\sum_j \sum_{p=0}^{d} \frac{\left[\frac{Q\,K^T}{\sqrt{d_k}}\right]_{m,j}^{p}}{d_k^{p/2}\, p!}} \tag{82}$$

and since $Q$, $K$ and $V$ are all linear transformations of the input data, we conclude that the attention can be arbitrarily approximated by a rationale function of degree $(2d + 1)/(2d)$. The degree $d$, however, may have to be impractically large in order reach proper accuracy, and therefore the approximation of the attention itself, albeit a member of PADRe, may not offer competitive performance in practice. Nonetheless, this shows that the standard attention falls within the PADRe framework as well.

Finally, let us highlight that such analysis is not limited to the softmax and the attention mechanism in particular, and may in fact be applied to any sufficiently regular (e.g., smooth) nonlinear functions (such as activations) as long as the input remains bounded in some interval. Upon making such a substitution, one finds that the resulting approximation is in fact a polynomial itself (given that all its building block are such). As part of some future work, we intend to generalize PADRe to this context as well.

