# OpenReview forum: "PADRe: A Unifying Polynomial Attention Drop-in Replacement for Efficient Vision Transformer"
_ICLR.cc/2025/Conference — ICLR 2025 Poster_

### Official Review · Reviewer_JNAG · 2024-10-29

**Soundness:** 3
**Presentation:** 3
**Contribution:** 3
**Rating:** 8
**Confidence:** 3

**Summary:**

This paper proposes a framework for drop in self attention replacement. This framework allows for building any polynomial time self-attention module, relies on polynomial approximates at the input, uses Hadamard product to introduce non-linearities. The framework at its current stage does not look into cross-attention and/or multi-modal attention although in principal it can be extended to cross-attention

**Strengths:**

- Framework that helps building alternative self-attention based transformer architecture
- advantageous over Flash attention

**Weaknesses:**

- The authors evaluated PADRe on DeiT. It might be a good idea to evaluate PADRe on ViT (Dosovitskiy et al.). This will demonstrate the impact of PADRe even more since ViT is widely used as an image encoder for many other models such as CLIP, Vision LLMs etc.
- Since you are evaluating on device, it would be good to show peak memory consumption, anything that shows energy efficiency improvement when running PADRe over standard self-attention

**Questions:**

- How does replacing self-attention with PADRe change the number of model parameters? Is there any increase or decrease? This will be helpful in understanding PADRe's impact on mobile devices since parameter count is directly correlated to storage, DRAM usage etc.

---

> ### Author Response · Authors · 2024-11-22
>
> We would like to thank Reviewer JNAG for the useful comments.
>
> Response to Comment 1:
>
> DeiT has the same architecture as ViT except for a distillation token. Essentially, DeiT is a more efficient training method on ImageNet using the same ViT architecture. In particular, the self-attention mechanism is exactly the same across DeiT and ViT.
>
> In addition, DETR and DSVT use standard transformer layers from ViT. As such, our DETR and DSVT experiments (Tables 2 & 3 in the paper) show that PADRe can effectively substitute ViT layers.
>
> Response to Comment 2:
>
> Considering one layer, given 4096 input tokens, PADRe’s peak memory consumption is 106MB while that of self-attention is 145MB, 37% higher than PADRe. Such advantages are expected to become even more prominent as one increases the number of tokens, given the linear scaling of PADRe.
>
> Response to Question 1
>
> As shown in Tables 1, 2 & 3 in the paper, PADRe with polynomial degree 2, i.e., PADRe-2, uses fewer parameters as compared to the standard self-attention. PADRe-3 requires slightly more parameters.

---

> > ### Comment · Reviewer_JNAG · 2024-11-23
> >
> > Thank you for clarifying. I will keep my score.

---

### Official Review · Reviewer_MgyZ · 2024-10-31

**Soundness:** 3
**Presentation:** 3
**Contribution:** 3
**Rating:** 8
**Confidence:** 3

**Summary:**

This paper proposes PADRe, a unifying framework for replacing the attention mechanism based on
polynomial approximants to provide an efficient, linear-complexity alternative to the
standard self-attention without sacrificing accuracy.

**Strengths:**

1. This paper addresses an important problem in vision transformer design.

2. This paper provides a theory-driven solution with a thorough mathematical justification.

**Weaknesses:**

1. How would the method work for large language models (LLMs), with much longer sequence lengths?

2. How to decide the optimal degree of PARDe?

**Questions:**

Please see the weaknesses above.

---

> ### Author Response · Authors · 2024-11-22
>
> We would like to thank Reviewer MgyZ for the useful comments.
>
> Response to Comment 1:
>
> PADRe can be applied to LLMs with minimal changes, e.g., changing standard convolution to causal convolution. As long as the input is a 1D sequence of tokens, PADRe can be used.
>
> We have performed an experiment on this. Specifically, we used a reduced Llama2 with ~300M parameters trained on SlimPajama as the baseline. Then, we replaced the self-attention with PADRe and retrained the model. On the test set, PADRe achieves almost the same cross-entropy as compared to the baseline (2.6 vs. 2.5). Note that this is just a preliminary result. More comprehensive experiments on larger language models will require significantly more time and resources, which we will consider as part of future work.
>
> Response to Comment 2:
>
> In Section 4.4 of the paper, our ablation study on the degree on ImageNet classification indicates diminishing returns when scaling the degree up. In theory, higher degrees offer stronger modeling power. One can select an appropriate degree based on the available computation budget.
>
> The optimal degree of PADRe, however, depends on the specific problem domain. Given the task and data, it is possible to find the optimal degree via existing neural architecture search methods, e.g., [1,2].
>
> [1] Gong, Chengyue, and Dilin Wang, “NasViT: Neural architecture search for efficient vision transformers with gradient conflict-aware supernet training,” ICLR 2022
>
> [2] Tan, Mingxing, and Quoc Le, "EfficientNet: Rethinking model scaling for convolutional neural networks," ICML 2019

---

### Official Review · Reviewer_o4xT · 2024-11-01

**Soundness:** 3
**Presentation:** 3
**Contribution:** 3
**Rating:** 6
**Confidence:** 4

**Summary:**

This paper introduces Polynomial Attention Drop-in Replacement (PADRe), a new method which aims to replace the conventional self-attention in Transformers. Specifically, PADRe replaces self-attention with polynomial approximants. Furthermore, it adopts mobile-friendly operations such as  Hadamard products to achieve a good balance between efficiency and performance.  The authors have provided rigorous proofs that the proposed PADRe is a general approach that covers previous Hyena, Mamba, SimA, etc. Extensive experiments have also demonstrated the effectiveness of their approach.

**Strengths:**

1. Good insight and strong theoretical proofs. Approximating self-attention with polynomial approximants sounds interesting. The authors have provided solid proofs that the proposed PADRe can cover other special cases in recent works.

2. Experiments have well demonstrated the effectiveness of their approach, especially considering the FLOPs reduction in Table 3 and Table 4.

3. The paper is in a good structure. Figures are clear as well.

**Weaknesses:**

1. The paper acknowledges that model performance begins to saturate (Lines 462-463) when the polynomial degree exceeds 3, suggesting potential numerical stability issues with higher-degree polynomials. Investigating more stable polynomial bases could be beneficial.

2. As the author already states, this paper lacks experiments on LLMs. It would be great if we can know the performance of PADRe on LLMs. This could be more attractive to the community than experimenting on ViTs.

3. In Table 1, the improvement over DeiT-Base is minor. Could it be that PADRe works more effectively for smaller Transformers?

**Questions:**

See the weakness.

---

> ### Author Response · Authors · 2024-11-22
>
> We would like to thank Reviewer o4xT for the useful comments.
>
> Response to Comment 1:
>
> Thanks for the useful suggestion! As mentioned in Section 5 of the paper, we will consider looking into more stable polynomial bases for higher degrees as part of future work.
>
> Response to Comment 2:
>
> We strongly believe that the current work demonstrates a clear potential of PADRe in the context of LLMs. Given that models like Mamba and Hyena are currently used at scale for LLMs and that they fall within the PADRe framework (as shown in the paper), PADRe will indeed prove successful when dealing with LLMs.
>
> To further concretize this, we have performed an experiment using our specific PADRe instance (Fig. 1 in the paper). Specifically, we used a reduced Llama2 with ~300M parameters trained on SlimPajama as the baseline. Then, we replaced the self-attention with PADRe and retrained the model. On the test set, PADRe achieves almost the same cross-entropy as compared to the baseline (2.6 vs. 2.5). Note that this is just a preliminary result. More comprehensive experiments on larger language models will require significantly more time and resources, which we will consider as part of future work.
>
> Response to Comment 3:
>
> We would like to remind the reviewer that the main goal of this paper is not to necessarily improve the accuracy of existing transformer-based architecture. Rather, our focus is on memory complexity, computational complexity and hardware-friendliness, all of which can suffer when using the standard attention. With regards to accuracy, we demonstrate that switching to PADRe has little impact and, as observed, can in fact be beneficial in some cases. As evidenced in Section 4.3 of the paper, PADRe’s actual computational performance is significantly superior.

---

> > ### Comment · Reviewer_o4xT · 2024-11-25
> > **Official Comment from Reviewer**
> >
> > Thanks for the authors' rebuttal. I have no more questions. It would be encouraging if the authors could explore the usage of PADRe in a broader applications in future works.

---

### Official Review · Reviewer_Lp3X · 2024-11-05

**Soundness:** 2
**Presentation:** 3
**Contribution:** 3
**Rating:** 5
**Confidence:** 4

**Summary:**

The paper proposes a novel approach to replace full-attention with a unified polynomial formulation that achieves linear complexity. The authors show that various efficient attention mechanisms can be seen as special cases within the PADRe framework. They also implement a specific instance of PADRe and observe a favorable trade-off between efficiency and accuracy, even with a polynomial degree as low as 3.

**Strengths:**

The unified formulation proposed in this paper is intriguing, as it establishes mathematical connections across different attention variants. I believe this polynomial-based approach has great potential for impact, offering a foundation for designing more efficient attention mechanisms in the future.

**Weaknesses:**

1): Figure 1 illustrates a specific design within the PADRe framework, but a straightforward baseline is missing. For example, for degree-
$d$ approximation, directly ensembling $d$ MLP-Mixers can be a strong baseline, and this approach can also be efficiently parallelized.

2): While the paper introduces a unified formulation, it remains unclear how readers can directly leverage it to enhance existing linear attention mechanisms. In Eq. (8), the coefficients $\pi_k$ are noted to have a complex dependency on parameters, specifically on the properties of the transformation matrices $A_i$, $B_i$, $C_i$ and $D_i$.  For example, in Eq. (82), a practical approximation formulation for the softmax operation can be more concrete.

3): The experimental results demonstrate comparable performance with previous methods, but the paper defines the transformation matrices as pointwise and 2D convolutions to mix channels and tokens, respectively. However, it is well-established that adding locality can boost ViT performance (e.g., LocalViT). To fully evaluate the PADRe's effectiveness, testing on NLP benchmarks and models—without adding any locality inductive bias—would provide a more compelling validation.

**Questions:**

See weakness.

---

> ### Author Response · Authors · 2024-11-22
>
> We would like to thank Reviewer Lp3X for the useful comments.
>
> Response to Comment 1:
>
> Indeed, d MLP mixers with Hadamard product-based output fusion falls within the polynomial approximation framework. However, MLP-Mixers is significantly less efficient given the fully-connected nature of the spatial mixing layers, which incurs quadratic computation and memory complexities w.r.t. the number of tokens. In contrast, PADRe provides a linear-complexity, hardware-friendly framework to substitute attention.
>
> Response to Comment 2:
>
> PADRe serves as a unified framework for designing efficient, hardware-friendly attention replacement. It also provides concrete guidance to enhance existing models, including linear attention networks and convolutional networks.
>
> In practice, A and B corresponds to token mixing and channel mixing before the Hadamard product (or any other efficient nonlinear mixing operation of choice), and C and D corresponds to token and channel mixing after. Token mixing can be efficiently implemented using a set of convolutional layers and channel mixing using a set of MLPs. These operations are well-optimized on modern hardware platforms. Also given that these are very commonly used operations, their hyperparameters such as numbers of token/channel mixing layers and kernel size can be readily optimized using established neural architecture search techniques.
>
> Note that, as shown in the paper (Table 1 and appendix), many existing linear attention schemes already fall within the PADRe framework, such as Castling-ViT and SimA. For improving simpler networks such as a convnet, one can ensemble d of them using a nonlinear fusion, as suggested in the reviewer’s first comment.
>
> Finally, the goal of this paper is not to approximate softmax. After softmax approximation with polynomials, the computation and memory complexities would still be quadratic.
>
> Response to Comment 3:
>
> We would like to clarify that we are not adding locality to boost ViT performance. Rather, PADRe provides an efficient, hardware-friendly approach to completely replace attention. We choose convolution to perform token mixing as it is computationally and memory efficient, and is also highly optimized on modern computation platforms.

---

### Author Response · Authors · 2024-11-22

We would like to thank the reviewers and ACs for their efforts in this review process. And we thank the reviewers for their constructive comments. We are encouraged that the reviewers find our work to have a great potential for impact, with strong mathematical foundation and extensive experiments demonstrating effectiveness of PADRe, and being more advantageous than FlashAttention.

We have addressed all the reviewers’ comments. Specifically, some reviewers asked about applying PADRe to LLMs. We note that existing models like Mamba and Hyena have already been used for LLMs and that they fall within PADRe framework, showing the successes of our PADRe framework for LLMs. We have also performed an experiment applying our specific PADRe instance to LLM, which further confirms the effectiveness of PADRe and indicates its great potential. More information is provided in our detailed responses below.

---

### Meta-Review · Area_Chair_YgPM · 2024-12-18

**Metareview:**

The PADRe paper introduces a polynomial attention mechanism as an efficient drop-in replacement for self-attention in vision transformers, offering hardware-friendly operations and maintaining accuracy while significantly reducing computational costs.

After rebuttal and discussions, this paper receives three positive ratings and one negative rating. After carefully  reviewing the paper and all the reviewers' comments and discussions, The AC agrees to accept the paper and strongly recommends incorporating the content from the rebuttal into the final version.

**Additional Comments On Reviewer Discussion:**

- Reviewer Lp3X acknowledges the intriguing unified formulation of PADRe and its potential impact, while noting the need for a straightforward baseline comparison and more concrete guidance on leveraging the framework for existing linear attention mechanisms. ,
- Reviewer o4xT commends the paper for its good insight, strong theoretical proofs, and well-structured experiments that demonstrate the effectiveness of PADRe, but suggests exploring more stable polynomial bases and conducting experiments on large language models (LLMs) to enhance the paper's appeal. ,
- Reviewer MgyZ sees the paper as addressing an important problem in vision transformer design with a theory-driven solution, yet raises questions about the application of PADRe to LLMs and the determination of the optimal degree for PADRe's polynomial approximants. ,
- Reviewer JNAG appreciates the framework's ability to build alternative self-attention modules and its advantage over Flash attention, but recommends evaluating PADRe on the original Vision Transformer (ViT) architecture and providing data on peak memory consumption to underscore energy efficiency improvements.

The rebuttal properly addresses most of the reviewers' initial concerns and reviewers keep the rating unchanged.

---

### Decision · Program_Chairs · 2025-01-22

Accept (Poster)